# Th17/Regulatory T-Cell Imbalance and Acute Kidney Injury in Patients with Sepsis

**DOI:** 10.3390/jcm11144027

**Published:** 2022-07-12

**Authors:** Xiao Zhou, Jingyi Yao, Jin Lin, Jingfeng Liu, Lei Dong, Meili Duan

**Affiliations:** 1Department of Critical Care Medicine, Beijing Friendship Hospital, Capital Medical University, Beijing 100050, China; zhouxiao@ccmu.edu.cn (X.Z.); jin0419@hotmail.com (J.L.); jingfengliu@ccmu.edu.cn (J.L.); wojiushidonglei@sina.com (L.D.); 2Experimental Center, Beijing Friendship Hospital, Capital Medical University, Beijing 100050, China; 15611350552@163.com

**Keywords:** acute kidney injury, Th17 cells (Th17), Th17/Treg imbalance, regulatory T cell (Treg), sepsis

## Abstract

To analyze the predictive value of the Th17/Treg ratio for renal injury in sepsis patients, a prospective observational study was conducted. Adult patients with sepsis were enrolled and divided into a sepsis-induced acute kidney injury (SAKI) group and a sepsis-without-AKI group. Logistic regression was used to analyze the independent predictors of SAKI, and the ROC curve was plotted to evaluate the predictive value of the Th17/Treg ratio for renal injury in patients with sepsis. A total of 124 patients were enrolled in this study, including 60 cases (48.39%) of SAKI. Patients who developed sepsis-induced acute kidney injury had a higher Th17/Treg ratio level compared to patients without it (0.11 [0.07, 0.28] versus 0.06 [0.05, 0.16], *p* < 0.05, respectively. The area under the receiver operating characteristic curve of the Th17/Treg ratio to predict sepsis-induced acute kidney injury was 0.669 (95% CI 0.574–0.763, *p* < 0.05). The Th17/Treg ratio was associated with SAKI (OR 1.15, 95%CI [1.06–1.24], *p* < 0.05, non-adjusted and R 1.12, 95%CI [1.00–1.25], *p* < 0.05, adjusted). The use of the Th17/Treg ratio improved the prediction performance of the prediction model of NAGL. The median Th17/Treg ratio significantly increased with the stratified KDIGO stage (*p* < 0.05). Th17/Treg imbalance was associated with occurrence of acute kidney injury and AKI severity in patients with sepsis. The Th17/Treg ratio could be a potential predictive marker of sepsis-induced acute kidney injury.

## 1. Introduction

Sepsis, defined as life-threatening organ dysfunction caused by a dysregulated host response to infection, is the leading cause of mortality in patients in the intensive care unit (ICU) worldwide [1]. Previous studies showed that from 2015 to 2016, the incidence of sepsis was 31.5 million, of which 19.4 million were severe sepsis, resulting in 5.3 million deaths in high-income countries worldwide [2]. By 2020, the numbers had increased to 48.9 million cases of sepsis, including 11 million sepsis-related deaths (19.7% of all deaths worldwide) [3]. The number of deaths related to sepsis even exceeds the number of deaths caused by bowel cancer and breast cancer [4]. Sepsis contributes to half of all in-hospital deaths in the US and is the leading cost for the US healthcare system [5,6]. Thus, sepsis is a major medical problem in critical care medicine needing more attention. Organ injury is a hallmark of sepsis, particularly acute kidney injury (AKI). According to the latest reports from Kidney Disease: Improving Global Outcomes (KDIGO), the incidence of AKI in critically ill patients was approximately 40%, and it has been reported in the literature that the incidence of AKI in critically ill patients can be as high as 70%, resulting in mortality up to 50% [7,8]. The most common cause of acute kidney injury in critically ill patients is sepsis. It is difficult to accurately define the exact time when acute kidney injury and sepsis occur in clinical practice. Complications that meet the diagnostic criteria of both sepsis and acute kidney injury and exclude other factors that may cause acute kidney injury are called sepsis-induced acute kidney injury (SAKI) [9]. SAKI accounted for approximately half of all acute kidney injury cases. The occurrence of acute kidney injury seriously affects the prognosis of patients. One-third of critically ill patients die within 90 days, and the mortality after 90 days is approximately 10%. Surviving patients often develop chronic renal insufficiency and require renal replacement therapy. Although renal replacement therapy and blood purification techniques in critically ill patients are mature, there is still a lack of effective interventions to reduce kidney injury and promote kidney repair. Therefore, the treatment of sepsis-induced acute kidney injury is a challenging and important clinical problem.

Immune disorder might play a vital role in the development of sepsis, and these disorders (immune excess or suppression) are closely related to T lymphocytes during both the early and late phases of sepsis [10]. Helper T cell 17 (Th17) and regulatory T cell (Treg) are two subsets of CD^4+^ T cells. They play different roles in the process of diseases and can positively or negatively modulate the immune response. They are followed by upstream cytokine formation, reaching a dynamic balance state. The Th17/Treg balance is regarded as a key factor in immune homeostasis. The imbalance of Th17/Treg cells has been confirmed to be associated with sepsis and various inflammatory diseases [11]. Recent studies have shown that the ratio of Th17/Treg cells is changed in patients with kidney injury and is more common in acute kidney injury than in chronic kidney disease. The development of SAKI is often accompanied by excessive release of IL-17 and tissue infiltration of Th17 cells [12]. Inhibition of Th17 differentiation and function can significantly alleviate acute kidney injury caused by the inflammatory response [13]. A number of animal experiments have confirmed that Tregs can play a role in renal protection in renal ischemia–reperfusion models and drug-induced renal injury models [14,15]. However, in a rat model of SAKI, conflicting conclusions regarding the role of Tregs in renal function have been obtained from different researchers [16,17]. In addition, there is a lack of clinical studies to confirm whether Tregs and Th17 cells play roles in the development of SAKI and renal recovery. Therefore, we designed this study to demonstrate the predictive value of Th17/Treg imbalance for sepsis-induced acute kidney injury.

## 2. Materials and Methods

### 2.1. Aim, Design and Setting

This was a prospective, observational and unicentric study. Patients hospitalized in the Department of Critical Care Medicine, Beijing Friendship Hospital, affiliated with Capital Medical University, from June 2021 to December 2021 were enrolled.

### 2.2. Participants

The inclusion criteria for patient enrollment were as follows: (1) age ≥ 18 years old; (2) sepsis that followed the Third International Consensus Definitions in 2016 [1]; and (3) signature for informed consent.

Exclusion criteria included (1) ICU hospital stay <24 h; (2) active connective tissue disease (e.g., rheumatoid arthritis, systemic lupus erythematosus); (3) pregnancy or breastfeeding patients; (4) previous hematological malignancies; (5) received radiotherapy or chemotherapy in the past 30 days; (6) received immunosuppressive drugs in the past 30 days or received continuous treatment of more than 10 mg of prednisolone per day (or other hormones at the same dose); (7) participated in clinical studies on related drugs or devices affecting immunity in the past 30 days; (8) withdrawal of therapy; and (9) any other circumstances that the researcher deemed unsuitable for participation in this study.

### 2.3. Data Collection

General clinical data and laboratory data of the patients were collected within 24 h of enrollment in sepsis patients without acute kidney injury (AKI) or diagnosis of SAKI.

General clinical data included sex, age, body mass index (BMI), history of past diseases, e.g., hypertension, diabetes, chronic kidney disease (CKD) stage 1 to 4, chronic cardiovascular disease, chronic lung disease, chronic liver disease, nervous system disease, rheumatic system disease and malignant tumor. A history of hypertension was defined as prior hypertension, treatment with antihypertensive medication or systolic blood pressure >140 mmHg or diastolic blood pressure >90 mmHg on admission. A history of diabetes was defined as previous or newly diagnosed diabetes, receiving oral hypoglycemic drugs or insulin injection hypoglycemic therapy, rapid random blood glucose greater than 7.0 mmol/L (126 mg/dL) or glycated hemoglobin >6.5%. A history of chronic kidney disease was defined as abnormal renal morphological or functional changes for more than 3 months. History of chronic cardiovascular disease, including heart disease and aortic and peripheral vascular disease. History of chronic lung disease including chronic obstructive pulmonary disease (COPD), cor pulmonale or pulmonary vascular disease. A history of chronic liver disease was defined as viral hepatitis, autoimmune hepatitis, metabolic or alcoholic liver disease and cirrhosis. History of neurological diseases, including cerebral hemorrhage, cerebral infarction and cerebral embolism. Chronic rheumatic diseases referred to patients who had previously suffered from rheumatic immune diseases and were not currently active or receiving long-term immunosuppressive therapy. The history of malignancy referred to patients who had or had not been treated for malignancy in the past or patients who had been admitted to the hospital for malignancy at this time, excluding patients who had received radiotherapy/chemotherapy in the past 30 days.

Laboratory data included serum creatinine (Cr), blood urea nitrogen (BUN), 24 h urine output, neutrophil gelatinase-associated lipocalin (NAGL), peripheral blood white blood cell (WBC), neutrophil granulocyte (GR) cell count, lymphocyte (LY) count, neutrophil/lymphocyte ratio (NLR), procalcitonin (PCT), lactic acid (Lac), oxygenation index (OI), T-lymphocyte ratio and CD^4+^ T-lymphocyte ratio. Serum creatinine (normal range 41–111 umol/L), blood urea nitrogen (normal range 2.6–7.5 mmol/L), peripheral white blood cell count (normal range 3.5–9.5 *×* 10^9^/L), neutrophil count (normal value range 1.8–6.3 *×* 10^9^/L), lymphocyte count (normal value range 1.1–3.2 *×* 10^9^/L), procalcitonin (normal value range), T-lymphocyte ratio (normal value range 58.6–83.1%) and CD^4+^ T-lymphocyte ratio (normal range 27.1–49.8%) were tested in the center laboratory within 24 h of enrollment. The 24 h urine output is the urine output of patients 24 h after enrollment. Lactic acid (normal value range 0.7–2.5 mmol/L) and oxygenation index were detected by a bedside blood gas instrument (GEM Premier 3500,Bedford, MA, USA) immediately after enrollment.

Information about infection, such as the site of infection, the amount of organ damage caused by infection and related treatments, was collected. Infection sites include respiratory system infections, urinary system infections, gastrointestinal infections, biliary tract infections, thoracic and abdominal infections, bloodstream infections, skin and soft tissue infections, central nervous system infections, mediastinal pericardium infections and other infections with unclear sites. The patient treatment measures included whether to use ventilators, vasoactive drugs, renal replacement therapy and blood transfusion therapy.

When the patients were enrolled, the researchers collected the relevant data and clinical indicators of the patients within 24 h to complete the Acute Physiology and Chronic Health Evaluation (APACHE) II and Sepsis-related Organ Failure Assessment (SOFA) score. If the patient’s condition was stable and they were transferred out of the ICU, the researchers will continue to complete clinical observation and collect clinical specimens and data. However, if the subsequent information could not be obtained, the patient was automatically eliminated. The length of hospital stay (LOS), ICU stay and ICU spending were also collected for all patients.

### 2.4. Blood Sampling and Measurements

Peripheral venous blood samples were collected immediately after enrollment. The peripheral venous blood (10 mL) of patients was extracted and placed into ethylene diamine tetraacetic acid (EDTA) anticoagulant tubes (5 mL) and blood collection tubes (5 mL) containing inert separation gel and coagulants. Blood samples were subjected to the Ficoll process to isolate peripheral mononuclear cells (PBMCs) from the whole-blood specimens within 2 h after drawing. The regulatory cell concentration was approximately 1 × 10^6^ cells/mL for flow cytometry, and the cells were labeled with a monoclonal antibody conjugated to fluorescein (CD3 mAb, CD4 mAb, IL-17 mAb, CD25 mAb, CD127 mAb) on a flow cytometry of InvitrogenTM AttuneTM NxT (Thermo Fisher Scientific, Waltham, MA, USA). The ratio of subcellular Th17/Treg (CD^3+^CD^4+^IL-17^+^/CD^3+^CD^4+^CD^25+^CD^127−^) of T lymphocytes in humans was determined using flow cytometry. Data were analyzed using FlowJo software (FlowJo, LLC. (Ashland, OR, USA)) (see Appendix A). Serum samples were prepared immediately by centrifugation of peripheral venous blood. Cytokines (Interleukin-10, Interleukin-17 and Tumor necrosis factor alpha) were measured by enzyme-linked immunosorbent assay (ELISA) according to the manufacturer’s guidelines (Biolegend, San Diego, CA, USA).

### 2.5. Group Analysis and Follow-Up

The primary endpoint for this study was occurrence of acute kidney injury, with 28-day mortality, hospital length of stay (LOS), ICU length of stay and expenses in the ICU as key secondary endpoints. All patients received standard treatment for severe sepsis or septic shock according to the Surviving Sepsis Campaign Guidelines [1]. They were divided into two groups according to the presence or lack of acute kidney injury. AKI was diagnosed as defined by 2012 KDIGO AKI clinical practice guidelines [18].

### 2.6. Statistical Analysis

Continuous variables conforming to a normal distribution are presented as mean ± SD. Independent samples *t*-tests were used for comparisons between two groups. Continuous variables that did not conform to a normal distribution are presented as median values and interquartile ranges (IQRs). The Mann–Whitney U test was used for comparisons between two groups. The Kruskal–Wallis test was applied for multivariate analysis. Categorical data were summarized using number (percentage), and were compared using the chi-square test. Binary univariate and multivariate logistic regression analyses were conducted for risk factor assessment. The area under the receiver operating characteristic (ROC) curve was calculated to evaluate the diagnostic and prognostic value of the tested parameters. The cutoff points were calculated by acquiring the best Youden index (sensitivity + specificity −1). Comparison of prediction performance was assessed using reclassification method. Sensitivity analyses were performed in subjects after exclusion of patients with urinary system and gastrointestinal infections. All reported probability values are two-tailed, and *p* < 0.05 was considered statistically significant. All missing data were included in the analysis. Analyses were performed using SPSS 25.0 software (IBM Corp., Armonk, NY, USA), and graphs were created by GraphPad Prism 8 (GraphPad, San Diego, CA, USA).

## 3. Results

### 3.1. Enrollment of Patients

Patients admitted to the Department of Critical Care Medicine, Beijing Friendship Hospital, affiliated with Capital Medical University, from June 2021 to December 2021 were screened and enrolled according to the inclusion and exclusion criteria. There were 124 patients enrolled, including 60 cases (48.39%) of sepsis-induced acute kidney injury. There were no COVID-19 patients in the cohort. The study flow chart is shown in Figure 1.

#### 3.1.1. Clinical Characteristics of Patients

Data obtained from 124 patients were analyzed. The baseline characteristics of the enrolled patients are displayed in Table 1. There were no significant differences in age, sex, BMI or comorbidities between the SAKI group and the sepsis-without-AKI groups, indicating that the patients in the two groups were comparable.

As seen in Table 1, we observed that the sepsis patients developed an average of three organ injuries. The average number of organ dysfunctions in the SAKI group was four, which was significantly higher than that in the patients with sepsis without AKI (4 vs. 2, *p* < 0.05). The SOFA scores were significantly higher in the SAKI group than in the sepsis-without-AKI group (8.00 vs. 4.50, *p* < 0.05). Septic shock was more frequent among SAKI patients than among sepsis patients without AKI (76.7% vs. 59.4%, *p* < 0.05). The groups did not differ in APACHE II score. NAGL, Cr, BUN, PCT, Lac and NLR were significantly higher in the SAKI group than in the non-AKI group, while urine output at 24 h was lower in the former group. This suggests that sepsis in patients with acute kidney injury was more severe than that in those without acute kidney injury. Therefore, the proportion of patients with SAKI using ventilators, vasoactive drugs and blood transfusions was higher.

We conducted an analysis per infectious site. Patients in the SAKI group were more likely to develop urinary tract and gastrointestinal infections, while those in the non-AKI group were more likely to have infections in the respiratory system, thoracic abdominal cavity and central nervous system.

#### 3.1.2. Th17/Treg Ratio Is Associated with the Occurrence of SAKI

We compared the results of total the T-lymphocytes ratio, CD^4+^ T-lymphocytes ratio, Th17 cells ratio, Treg cells ratio and related cytokines in the peripheral blood of the two groups. The proportions of total T lymphocytes, CD^4+^ T lymphocytes and Treg cells were not significantly different between the two groups in our study. The proportion of Th17 cells in the peripheral blood of patients in the SAKI group was significantly higher than that in the control group (0.25 vs. 0.07, *p* < 0.05). Thus, compared with the sepsis-without-AKI group, the Treg/Th17 ratio in the SAKI group showed a significant increase (0.11 vs. 0.06, *p* < 0.05) (see Table 1).

Cytokine concentrations in the peripheral blood of patients were also measured. TNF-α is a primary inflammatory factor that is released during the early stages of inflammation. IL-10 and IL-17 are the major cytokines of Treg and Th17 cells. The concentrations of IL-10 and IL-17 were significantly higher in the SAKI group than in the sepsis-without-AKI group. The concentrations of TNF-α revealed no significant difference between the two groups. Changes in the concentration of IL-17 and the Th17 cell ratio were consistent. However, changes in IL-10 concentration were not always consistent with the trend in the proportion of Treg cells (see Figure 2).

Table 2 shows univariate and multivariate analyses of risk factors for the occurrence of SAKI. Regression analysis was performed with the occurrence of SAKI as the dependent variable and with the factors with significant differences in the univariate analysis as the independent variables. Variables that had a *p* value of less than 0.05 according to the univariable binary logistic regression analysis included SOFA, occurrence of septic shock, infection of respiratory system, infection of urinary system, infection of thoracic and abdominal cavity, number of organ dysfunctions caused by infection, use of vasoactive drugs, use of blood transfusion, urine output, NAGL, Cr, BUN, PCT, Lac, Th17 cells ratio, concentration of IL-10, concentration of IL-17 and Th17/Treg ratio. Ultimately, SOFA, NAGL, PCT, concentration of IL-10, concentration of IL-17 and Th17/Treg ratio were included in the multivariate logistic regression analysis. Independent predictors of SAKI were identified at *p* < 0.05. Multivariate regression analysis revealed that the SOFA, NAGL, PCT and Th17/Treg ratio were independent risk factors for the occurrence of acute kidney injury in sepsis patients (see Figure 3).

ROC curves were plotted to further compare the role of NAGL and Th17/Treg ratio in predicting the occurrence of SAKI and to seek the cutoff value of the Th17/Treg ratio as a predictor of SAKI (see Figure 4). The AUC demonstrated that the Th17/Treg ratio was 0.669 (95% CI 0.574–0.763, *p* < 0.05) and the NAGL was 0.769 (95% CI 0.679–0.858, *p* < 0.05). The AUC for the Th17/Treg ratio in combination with the NAGL was 0.798 (95% CI 0.716–0.879, *p* < 0.05), which was higher than that for the NAGL or Th17/Treg ratio alone for predicting the occurrence of AKI in patients with sepsis. The cutoff value of the Th17/Treg ratio for predicting the occurrence of AKI was 0.06967. When the Th17/Treg ratio was >0.06967, the sensitivity of predicting the occurrence of SAKI was 0.800, and the specificity was 0.531.

Next, the patients were regrouped by the cutoff value of the Th17/Treg ratio into the high-Th17/Treg-ratio group (Th17/Treg ratio > 0.06967) or low-Th17/Treg-ratio group (Th17/Treg ratio ≤ 0.06967). The incidence of AKI was significantly higher in the high-Th17/Treg-ratio group than in the low-Th17/Treg-ratio group (61.5% vs. 26.1%, *p* < 0.05) (see Figure 5). Then, univariate and multivariate logistic regression were performed with the occurrence of SAKI as the dependent variable and with the binary variable high-Th17/Treg-ratio as the independent variable (see Table 3). A high Th17/Treg ratio was an independent risk factor for the occurrence of SAKI (OR = 8.16, 95% CI 1.89–35.14, *p* < 0.05). This finding verifies the results of the previous regression. Th17/Treg imbalance can be used as an independent risk factor for predicting the occurrence of SAKI.

Sensitivity analysis was carried out for the subjects after exclusion of patients with gastrointestinal infections, and we obtained the same conclusion: Th17/Treg ratio was an independent risk factor for the occurrence of SAKI (see Appendix A).

#### 3.1.3. General Outcomes of Patients

The 28-day mortality, LOS and expenses in the ICU were higher in either the SAKI group or high-Th17/Treg-ratio group, but the difference was not statistically significant (see Figure 5 and Figure 6).

#### 3.1.4. Increased Th17/Treg Ratio in Patients with SAKI

Sixty-four cases (51.61%) of sepsis patients had no AKI with a median Th17/Treg ratio of 0.06 (0.05, 0.16). Sixty cases (48.39%) of sepsis patients developed acute kidney injury. A total of 8 cases (6.4%) had stage 1 AKI with a median Th17/Treg ratio of 0.05 (0.04, 0.23), 20 cases (16.13%) had stage 2 AKI with a median Th17/Treg ratio of 0.11 (0.06, 0.36), and 32 cases (25.81%) had stage 3 AKI with a median Th17/Treg ratio of 0.13 (0.08, 0.25) (see Figure 7). The median Th17/Treg ratio significantly increased with stratified KDIGO stage (*p* < 0.05). Patients with stage 3 AKI had a high Th17 cell ratio and concentration of cytokines (IL-10, IL-17, TNF-α), and yet a low Treg cell ratio.

In the subgroup analysis of patients with sepsis-induced acute kidney injury, the AKI stage 3 group of patients had medium rises in the Th17/Treg ratio levels compared to that of the AKI stage 1 and 2 group with no significant differences (see Appendix A).

## 4. Discussion

The main finding of our study was that the Th17/Treg ratio, SOFA, NAGL and PCT could be independent risk factors for AKI in sepsis patients. In this study, an elevated proportion of Th17 cells and an imbalance between Treg and Th17 cells were observed in patients with SAKI. The proportion of Treg cells did not change significantly in sepsis patients with AKI or without. This result suggests that there was an imbalance in the Th17/Treg ratio in the process of acute kidney injury in patients with sepsis, and the Th17/Treg balance axis trended toward the Th17 cell lineage. As effectors, the concentrations of IL-10 and IL-17 were significantly higher in the SAKI group than in the sepsis-without-AKI group. Changes in the concentration of IL-17 and the Th17 cell ratio were consistent. Changes in IL-10 concentration were not consistent with the trend in the proportion of Treg cells. The reason for this may be that IL-10 can be secreted by other immune cells, such as Tr1 and Th3 cells, in addition to Treg cells. The concentrations of TNF-α revealed no significant difference between the two groups. TNF-α is a major inflammatory factor in the early stages of infection and has characteristics including earlier release and quicker recovery. Sepsis patients admitted to the ICU were not always in the early stages of disease. This may be the reason why there was no difference in TNF-α in the sepsis patients with and without AKI.

Under physiological conditions, Th17 and Treg cells are in a dynamic equilibrium. A variety of autoimmune diseases are caused when the Th17/Treg balance is skewed [19,20,21,22,23,24,25]. Current studies have confirmed that the imbalance of Th17/Treg cells is crucial during the immunopathogenesis of sepsis. In this study, we confirmed that Th17/Treg imbalance was associated with the occurrence of acute kidney injury in patients with sepsis. Currently, NGAL is widely recognized as a biomarker for acute kidney injury [26,27,28]. Our research also confirmed this and found that the combination of the Th17/Treg ratio and NAGL enhanced predictive power of acute kidney injury in sepsis patients. Subgroup analysis showed than for patients with sepsis-induced acute kidney injury, the Th17/Treg ratio increased with stratified KDIGO stage, but NAGL did not exhibit such a trend. This highlights a potential role of the Th17/Treg ratio in predicting not only the occurrence of SAKI but also the severity of AKI. This conclusion could be helpful for early identification and diagnosis of SAKI.

We analyzed the likely mechanism through the literature. Studies on animal models of drug-induced and ischemia–reperfusion acute kidney injury revealed that Treg cells can exert a renal protective effect by modulating proinflammatory cytokines of T-cell subpopulations [15,17,29,30,31]. However, in a mouse model of sepsis-induced acute kidney injury, Lee et al. found that depletion of regulatory T cells before cecal ligation and puncture can produce renal protection; inhibition of IL-10 can prevent the development of sepsis-related acute kidney injury, which was contrary to the conclusions of previous studies [16]. Th17 cells can increase the immune response mainly through the secretion of IL-17. Previous studies of different animal models showed that IL-17 can act on several immune cells as well as innate cells of the kidney [32]. It stimulates innate cells of the kidney to produce chemokines, which, in turn, recruit inflammatory cells. Studies on animal models of ischemia–reperfusion kidney injury suggested that IL-17 may participate in the development and progression of renal fibrosis through neutrophil recruitment [33] or by stimulating renal tubular epithelial cells and podocyte expression of TGF-β and extracellular matrix (ECM) proteins [34]. At the same time, IL-17 acts directly on the kidney, downregulates E-cadherin expression, affects the links between renal tubular epithelial cells and induces podocyte apoptosis and cytoskeletal damage.

In the study, the SOFA score, the proportion of septic shock and the number of organ dysfunctions caused by infection were significantly higher in the SAKI group than in the sepsis-without-AKI group. This suggests that sepsis patients with acute kidney injury were in a more severe condition than sepsis patients without AKI, which is in accordance with previous reports [35]. Accordingly, the proportions of vasoactive drugs and blood transfusions used in sepsis-induced acute kidney injury patients were higher. The SOFA score actually reflects the damage to organ function, which is positively correlated with the number and degree of organ dysfunction. Septic shock and kidney injury both contribute to the increase in the SOFA score. In the study, we did not separately calculate the SOFA score other than for renal function. It is currently agreed that septic shock leads to decreased renal perfusion, which can easily lead to renal damage. Therefore, it was reasonable that the proportion of septic shock was higher in the SAKI patient group. In this study, the incidence of acute kidney injury in sepsis patients was 48.39%, which is consistent with previous literature studies. Nevertheless, the 28-day mortality of sepsis patients was 16.1%, which is lower than the mortality reported in the literature [3]. This may be related to the fact that most sepsis patients in our center came from surgery departments. Timely surgical operation removed the foci of infections and improved the prognosis of patients. The 28-day mortality, LOS, in ICU length of stay and expenses in the ICU were higher in the SAKI group and the high-Th17/Treg-ratio group than in the control groups, but the differences were not significant. This may be related to the small sample size.

The distribution of the infection sites showed that patients in the SAKI group were more likely to develop urinary tract and gastrointestinal infections, while those in the non-AKI group were more likely to have infections in the respiratory system, thoracic abdominal cavity and central nervous system. It seemed that intra-abdominal organ infections were more likely to lead to the occurrence of SAKI, but whether this phenomenon was related to the source of patients in this research center was unknown. A multicenter study with a large sample size can be carried out in the future. In addition, Th17 cells have been demonstrated to significantly contribute to the inflammation in the gastrointestinal tract [36]. Patients in the SAKI group had a higher incidence of gastrointestinal infections. In order to rule out the possible effect of gastrointestinal infection on the Th17/Treg ratio, analysis of the subjects after exclusion of patients with gastrointestinal infections confirmed our original conclusions.

There were several limitations in the study. First, the small sample size and the single-center study design limited the generalizability and external validity of the results. However, the results of the study are sufficient to observe the correlation between the imbalance of Th17/Treg and the occurrence of acute kidney injury in sepsis. Second, we only investigated the imbalance of Th17/Treg in the peripheral blood of patients. Ideally, samples should be taken from kidney tissue. However, it is not feasible to obtain the appropriate kidney tissue specimens in clinical trials, especially in critically ill patients. The results of the study confirm that it was safe and feasible to analyze the correlation with the occurrence of sepsis-induced AKI and the change in the Th17/Treg ratio in peripheral blood samples. Third, most of the patients had developed sepsis-induced acute kidney injury at the time of enrollment, and the courses of the disease were not entirely consistent. The enrolled patients were diagnosed with acute kidney injury according to the 2012 KDIGO guidelines regarding changes in creatinine or urine output, rather than the injury of kidney tissue. Still, we detected NAGL as an early marker of renal injury to evaluate the predictive ability of the Th17/Treg ratio for early kidney injury. Fourth, novel biomarkers for kidney injury have emerged in recent years. Urinary insulin-like growth factor-binding protein (IGFBP-7) and tissue inhibitor of metalloproteinase (TIMP-2) have been proposed as predictive tools for AKI detection, which showed a stronger predictive power than NAGL. The Nephrocheck^®^ test is a single-use cartridge designed to measure the concentrations of two novel cell-cycle arrest biomarkers of acute kidney injury [27,37]. Further studies have confirmed that using Nephrocheck to predict the risk of acute kidney injury was related to urine flow. As a result, for patients with oliguria or anuria, the predictive ability needs to be judged comprehensively [38]. Unfortunately, we did not collect patients’ urine, so we could not detect urinary TIMP-2 or IGFBP-7 in the study. Kashani K’s study demonstrated that NAGL was weaker than NephroCheck and urinary TIMP-2/IGFBP7, but stronger than plasma cystatin C and urine KIM-1 in predicting acute kidney injury [27]. We detected serum NAGL as a marker of early kidney injury in the study and obtained positive results. The latest literature, such as Liu’s study [28] and Jia’s study [39], analyzed NAGL as a predictive biomarker for AKI. Therefore, we think it is feasible to use NAGL as a biomarker for AKI. We plan to enrich the detection of biomarkers for acute kidney injury in further studies. Finally, a correlation between the imbalance of Th17/Treg and acute kidney injury in sepsis was observed in the present study. The mechanism remains to be elucidated. This is also a major part of our future research.

Altogether, these data suggest that the Th17/Treg ratio is strongly associated with SAKI and could indicate the severity stratification for AKI. Therefore, regulation of Th17/Treg balance appears to be a promising strategy to improve outcomes in sepsis patients with kidney damage.

## 5. Conclusions

Th17/Treg imbalance is associated with occurrence of acute kidney injury and AKI severity in patients with sepsis. The Th17/Treg ratio could be a potential predictive marker of sepsis-induced acute kidney injury.

## Figures and Tables

**Figure 1 jcm-11-04027-f001:**
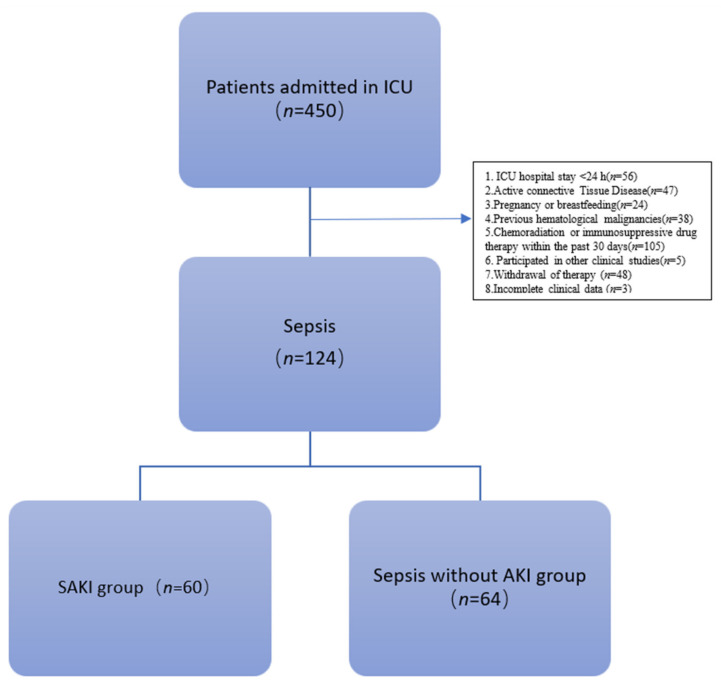
Study flow chart. SAKI: sepsis-induced acute kidney injury; AKI, acute kidney injury; ICU, intensive care unit.

**Figure 2 jcm-11-04027-f002:**
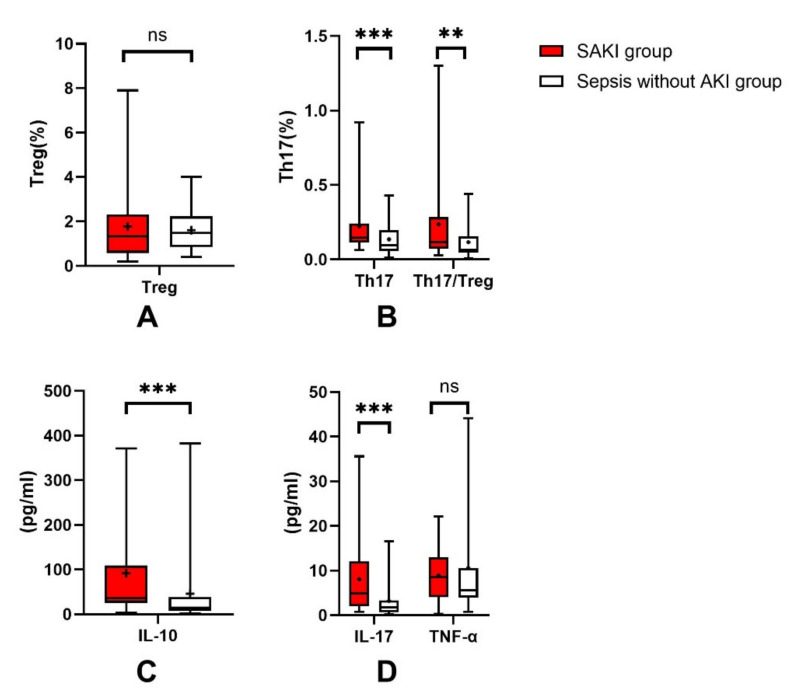
Treg ratio for SAKI group and sepsis-without-AKI group (**A**). Th17 ratio and Th17/Treg ratio for SAKI group and sepsis-without-AKI group (**B**). IL-10 level for SAKI group and sepsis-without-AKI group (**C**). IL-17 and TNF-α level for SAKI group and sepsis-without-AKI group (**D**). SAKI: sepsis-induced acute kidney injury; Treg, regulatory T cell; Th17, helper T cell 17; IL-10, Interleukin-10; IL-17, Interleukin-17; TNF-α, Tumor necrosis factor alpha; AKI, acute kidney injury. *p* value: ns > 0.05; ** 0.01–0.001; ***< 0.001.

**Figure 3 jcm-11-04027-f003:**
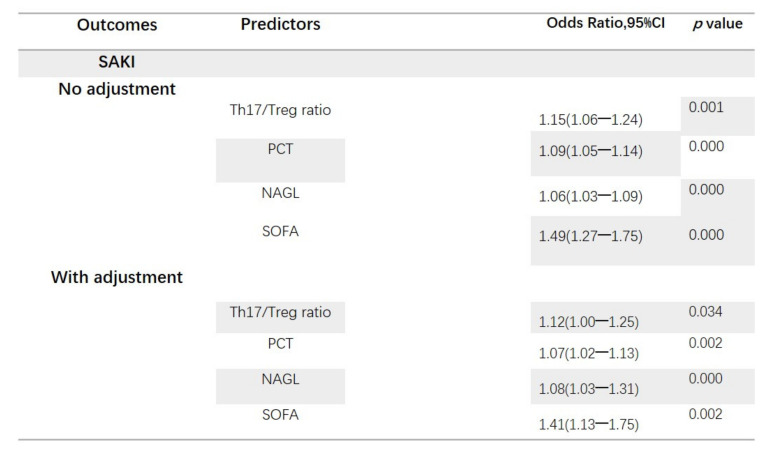
Th17/Treg ratio, PCT, NAGL and SOFA after adjustment association with the occurrence of sepsis-induced acute kidney injury. OR for continuous variables (Th17/Treg ratio, PCT, NAGL, SOFA). OR, odds ratio; CI, confidence interval; SAKI, sepsis-induced acute kidney injury; Treg, regulatory T cell; Th17, helper T cell 17; SOFA, Sequential Organ Failure Assessment; NAGL, neutrophil gelatinase-associated lipocalin; PCT, procalcitonin.

**Figure 4 jcm-11-04027-f004:**
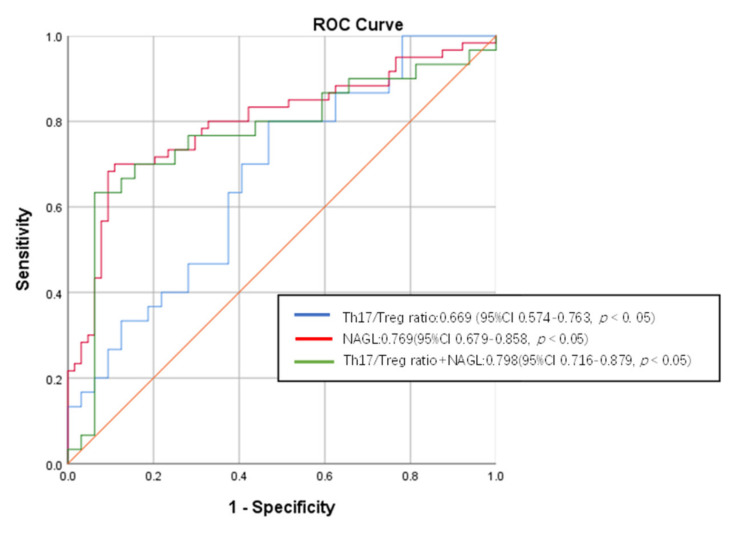
ROC curve of Th17/Treg ratio and NAGL prediction performance for SAKI. SAKI, sepsis-induced acute kidney injury; ROC, receiver operating characteristics; CI, confidence interval; Treg, regulatory T cell; Th17, helper T cell 17; NAGL, neutrophil gelatinase-associated lipocalin.

**Figure 5 jcm-11-04027-f005:**
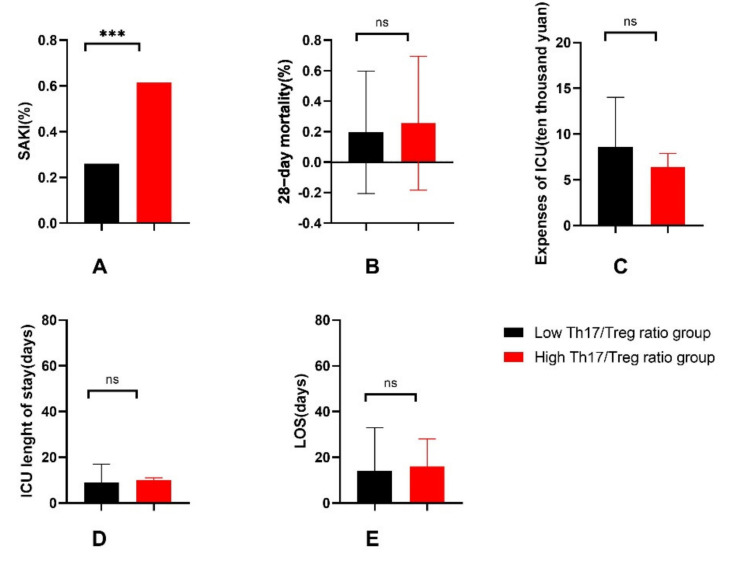
General outcomes between the high-Th17/Treg-ratio (>0.06967) group and the low-Th17/Treg-ratio (≤0.06967) group, including the comparison of incidence of SAKI (**A**), 28-day mortality (**B**), expenses of ICU (**C**), ICU length of stay (**D**), LOS (**E**). SAKI, sepsis-induced acute kidney injury; LOS, hospital length of stay; ICU, intensive care unit. *p* value: ns > 0.05; ***< 0.001.

**Figure 6 jcm-11-04027-f006:**
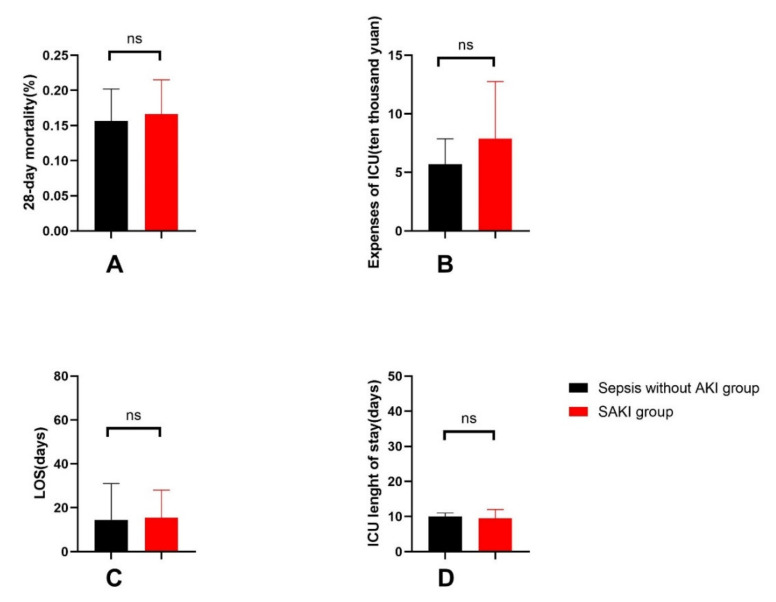
General outcomes between the SAKI group and the sepsis-without-AKI group, including the comparison of 28-day mortality (**A**), expenses of ICU (**B**), LOS (**C**), ICU length of stay (**D**). SAKI, sepsis-induced acute kidney injury; LOS, hospital length of stay; ICU, intensive care unit. *p* value: ns > 0.05.

**Figure 7 jcm-11-04027-f007:**
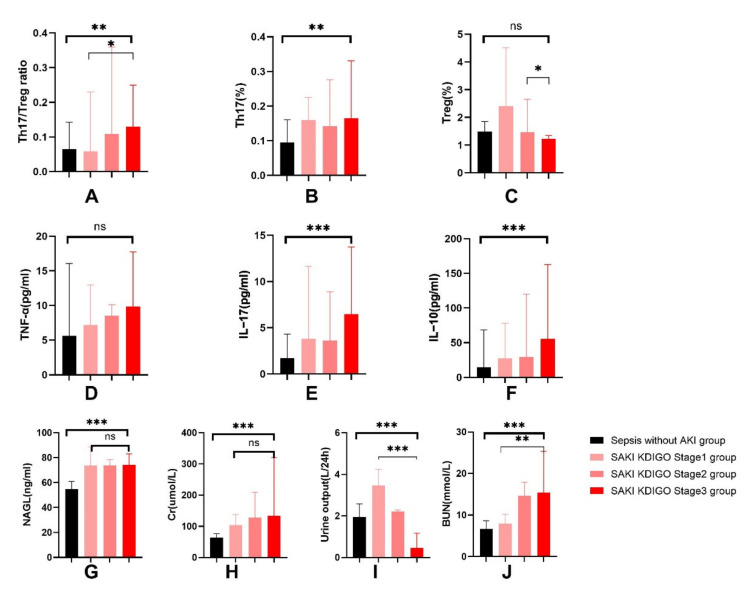
Changes in immune and renal function indicators in sepsis-without-AKI group and different severity of SAKI groups, including Th17/Treg ratio (**A**), Th17 cell ratio (**B**), Treg cell ratio (**C**), TNF-α (**D**), IL-17 (**E**), IL-10 (**F**), NAGL (**G**), Cr (**H**), urine output (**I**) and BUN (**J**). SAKI, sepsis-induced acute kidney injury; NAGL, neutrophil gelatinase-associated lipocalin; Cr, serum creatinine; BUN, blood urea nitrogen; Th17, helper T cell 17; Treg, regulatory T cell; IL-10, Interleukin-10; IL-17, Interleukin-17; TNF-α, Tumor necrosis factor alpha. *p* value: ns > 0.05, * 0.05–0.01; ** 0.01–0.001; *** < 0.001.

**Table 1 jcm-11-04027-t001:** Comparison of clinical characteristics between SAKI and sepsis-without-AKI groups.

	Total (*n* = 124)	SAKI Group (*n* = 60)	Sepsis-without-AKI Group (*n* = 64)	Test Value	*p* Value
Male [*n* (%)]	70 (56.5)	38 (63.3)	32 (50.0)	2.239	0.135
Age—year (median [Q1, Q2])	67.50 (55.00, 75.00)	67.50 (54.00, 76.00)	67.50 (56.25, 74.50)	−0.130	0.896
BMI—kg/m^2^ (mean ± SD)	23.70 ± 4.79	23.33 ± 4.76	23.95 ± 4.84	0.586	0.559
Comorbidities					
Hypertension [*n* (%)]	74 (59.7)	40 (66.7)	34 (53.1)	2.360	0.124
Diabetes [*n* (%)]	36 (29.0)	22 (36.7)	14 (21.9)	3.289	0.070
CKD [*n* (%)]	20 (16.1)	10 (16.7)	10 (15.6)	0.025	0.875
Chronic cardiovascular disease [*n* (%)]	50 (40.3)	26 (43.3)	24 (37.5)	0.438	0.508
Chronic lung disease [*n* (%)]	16 (12.9)	10 (16.7)	6 (9.4)	1.465	0.226
Chronic liver disease [*n* (%)]	8 (6.5)	2 (3.3)	6 (9.4)	1.873	0.171
Nervous system disease [*n* (%)]	36 (29.0)	14 (23.3)	22 (34.4)	1.832	0.176
Rheumatic system disease [*n* (%)]	4 (3.2)	2 (3.3)	2 (3.1)	0.004	0.948
Malignant tumor [*n* (%)]	38 (30.6)	16 (26.7)	22 (34.4)	0.866	0.352
Physiological parameters					
SOFA score (median [Q1, Q2])	7.00 (4.00, 8.00)	8.00 (6.00, 11.00)	4.50 (3.00, 7.00)	−5.554	0.000 ***
APACHEII score (median [Q1, Q2])	20.00 (15.00, 25.00)	20.50 (15.00, 25.00)	20.00 (15.00, 25.00)	−0.271	0.787
Septic shock [*n* (%)]	84 (67.7)	46 (76.7)	38 (59.4)	4.237	0.040 *
Numbers of organs with injuries caused by infection (median [Q1, Q2])	3 (2, 5)	4 (3, 5)	2 (1, 3)	−6.138	0.000 ***
Site of infection					
respiratory system [*n* (%)]	58 (46.8)	22 (36.7)	36 (56.3)	4.770	0.029 *
urinary system [*n* (%)]	20 (16.1)	18 (30.0)	2 (3.1)	16.534	0.000 ***
gastrointestinal [*n* (%)]	4 (3.2)	4 (6.7)	0 (0)	4.409	0.036 *
biliary tract [*n* (%)]	6 (4.8)	4 (6.7)	2 (3.1)	0.844	0.358
Thoracic and abdominal cavity [*n* (%)]	84 (67.7)	32 (53.5)	52 (81.3)	11.044	0.001 **
bloodstream [*n* (%)]	4 (3.2)	2 (3.3)	2 (3.1)	0.004	0.948
skin and soft tissue [*n* (%)]	2 (1.6)	0 (0)	2 (3.1)	1.906	0.167
central nervous system [*n* (%)]	4 (3.2)	0 (0)	4 (6.3)	3.875	0.049 *
NAGL—ng/ mL(median [Q1, Q2])	60.021 (53.190, 75.104)	74.101 (60.013, 80.943)	54.676 (51.057, 61.933)	−5.161	0.000 ***
Urine output—L/24 h (median [Q1, Q2])	1.81 (1.15, 2.47)	1.21 (0.38, 2.21)	2.02 (1.37, 2.67)	−3.120	0.002 **
Cr—umol/L (median [Q1, Q2])	80.55 (59.80, 128.10)	130.55 (100.40, 229.65)	63.70 (50.28, 74.13)	−8.081	0.000 ***
BUN—mmol/L (median [Q1, Q2])	9.92 (5.89, 15.12)	12.90 (10.22, 21.89)	6.34 (4.38, 8.42)	−5.821	0.000 ***
PCT—ng/ mL(median [Q1, Q2])	2.91 (0.70, 19.40)	19.07 (3.31, 33.84)	1.31 (0.55, 4.44)	−5.594	0.000 ***
Lac—mmol/L (median [Q1, Q2])	1.50 (1.10, 2.00)	1.80 (1.15, 3.08)	1.40 (1.00, 1.83)	−3.054	0.002 **
OI (median [Q1, Q2])	248.00 (180.00, 342.00)	245.00 (184.50, 368.75)	257.00 (178.25, 325.75)	−0.760	0.447
WBC—x10^9^/L (median [Q1, Q2])	11.57 (8.17, 16.77)	12.49 (7.99, 22.18)	12.02 (8.28, 14.59)	−1.180	0.238
GR—x10^9^/L (median [Q1, Q2])	10.29 (6.84, 15.09)	11.45 (7.08, 20.03)	10.41 (6.78, 12.92)	−1.600	0.110
LY—x10^9^/L (median [Q1, Q2])	0.72 (0.50, 1.08)	0.72 (0.47, 1.28)	0.87 (0.54, 1.13)	−0.720	0.471
NLR (median [Q1, Q2])	13.31 (8.72, 20.56)	16.65 (0.24, 32.78)	11.80 (7.71, 18.40)	−2.680	0.007 **
Immune indexes					
T-lymphocyte ratio—% (median [Q1, Q2])	69.01 (62.58, 78.81)	68.81 (55.49, 75.65)	70.59 (62.91, 83.97)	−1.301	0.193
CD^4+^ T-lymphocyte ratio—% (median [Q1, Q2])	40.50 (27.72, 50.74)	39.72 (27.88, 49.54)	42.79 (26.69, 51.18)	−0.410	0.682
Treg cell ratio—% (median [Q1, Q2])	1.40 (0.77, 2.25)	1.34 (0.59, 2.32)	1.49 (0.86, 2.23)	−0.510	0.610
Th17 cell ratio—% (median [Q1, Q2])	0.13 (0.08, 0.23)	0.15 (0.11, 0.24)	0.09 (0.06, 0.19)	−3.511	0.000 ***
Th17/Treg ratio (median [Q1, Q2])	0.10 (0.05, 0.21)	0.11 (0.07, 0.28)	0.06 (0.05, 0.16)	−3.240	0.001 **
IL10—pg/ mL(median [Q1, Q2])	25.55 (10.70, 63.84)	35.55 (25.46, 109.01)	14.48 (7.41, 37.94)	−4.541	0.000 ***
IL17—pg/ mL(median [Q1, Q2])	2.70 (1.03, 6.48)	4.87 (2.03, 12.02)	1.71 (0.70, 3.32)	−4.399	0.000 ***
TNF-α—pg/ mL(median [Q1, Q2])	6.48 (4.03, 11.16)	8.51 (4.05, 12.98)	5.59 (3.95, 10.55)	−0.403	0.687
General outcomes					
28-day mortality [*n* (%)]	20 (16.1)	10 (16.7)	10 (15.6)	0.025	0.875
Hospital length of stay in days (median [Q1, Q2])	15.00 (10.00, 35.00)	15.50 (8.00, 32.00)	14.50 (10.00, 39.50)	−0.671	0.502
ICU length of stay in days (median [Q1, Q2])	10.00 (6.00, 17.00)	9.50 (7.00, 17.00)	10.00 (5.00, 17.50)	−0.802	0.423
Expenses in ICU—CNY ten thousand (median [Q1, Q2])	7.06 (3.47, 14.08)	7.90 (4.04, 17.84)	5.69 (3.46, 13.64)	−1.430	0.153
Treatments during hospitalization					
ventilator [*n* (%)]	72 (58.1)	36 (60.0)	36 (56.3)	0.179	0.672
vasoactive drugs [*n* (%)]	78 (62.9)	44 (73.3)	34 (53.1)	5.420	0.020
blood transfusion [*n* (%)]	38 (30.6)	28 (46.7)	10 (15.6)	14.040	0.000 ***

BMI, body mass index; CKD, chronic kidney diseases; SOFA, sequential organ failure assessment; APACHE II, Acute Physiology and Chronic Health Evaluation 2; NAGL, neutrophil gelatinase-associated lipocalin; Cr, serum creatinine; BUN, blood urea nitrogen; PCT, procalcitonin; Lac, lactic acid; OI, oxygenation index; WBC, white blood cell; GR, neutrophil granulocyte; LY, lymphocyte; NLR, neutrophil/lymphocyte ratio; Th17, helper T cell 17; Treg, regulatory T cell; IL-10, Interleukin-10; IL-17, Interleukin-17; TNF-α, Tumor necrosis factor alpha; AKI, acute kidney injury; ICU, intensive care unit. *p* value: * 0.05–0.01; ** 0.01–0.001; *** < 0.001.

**Table 2 jcm-11-04027-t002:** Risk factor analysis for the occurrence of SAKI.

	Univariable Logistic Regression		Multivariable Logistic Regression	
	Odds Ratio (95% CI)	*p* Value	Odds Ratio (95% CI)	*p* Value
SOFA	1.49 (1.27–1.75)	0.000 ***	1.41 (1.13–1.75)	0.002 **
Septic shock	0.45 (0.20–0.97)	0.042 *		
Infection of respiratory system	2.22 (1.08–4.57)	0.030 *		
Infection of urinary system	0.08 (0.02–0.34)	0.001 **		
Infection of gastrointestinal	-	0.999		
Infection of thoracic and abdominal cavity	3.79 (1.69–8.50)	0.001 **		
Infection of central nervous system	-	0.999		
Number of organ dysfunctions caused by infection	2.44 (1.78–3.36)	0.000 ***		
Use of vasoactive drugs	0.41 (0.19–0.88)	0.021 *		
Use of blood transfusion	0.21 (0.09–0.49)	0.000 ***		
NAGL	1.06 (1.03–1.09)	0.000 ***	1.08 (1.03–1.31)	0.000 ***
Urine output	0.58 (0.41–0.82)	0.002 **		
Cr	1.04 (1.02–1.060)	0.000 ***		
BUN	1.26 (1.15–1.39)	0.000 ***		
PCT	1.09 (1.05–1.14)	0.000 ***	1.07 (1.02–1.13)	0.002 **
Lac	1.33 (1.04–1.72)	0.022 *		
NLR	1.01 (0.99–1.02)	0.249		
Th17	95.39 (3.45–2600.43)	0.007 **		
IL-10	1.01 (1.00–1.01)	0.021 *	0.99 (0.98–1.00)	0.266
IL-17	1.15 (1.06–1.24)	0.006 **	1.12 (1.00–1.25)	0.054
Th17/Treg ratio	46.63 (2.93–741.578)	0.001 **	144.99 (1.20–17,383.77)	0.034 *

SAKI, sepsis-induced acute kidney injury; CI, confidence interval; SOFA, Sequential Organ Failure Assessment; NAGL, neutrophil gelatinase-associated lipocalin; Cr, serum creatinine; BUN, blood urea nitrogen; PCT, procalcitonin; Lac, lactic acid; NLR, neutrophil/lymphocyte ratio; Th17, helper T cell 17; IL-10, Interleukin-10; IL-17, Interleukin-17. *p* value: * 0.05–0.01; ** 0.01–0.001; *** < 0.001.

**Table 3 jcm-11-04027-t003:** Multivariable logistic regression of reclassification by cutoff value of Th17/Treg ratio.

	Odds Ratio (95% CI)	*p* Value
High Th17/Treg ratio	8.16 (1.89–35.14)	0.005 **
SOFA	1.36 (1.10–1.70)	0.005 **
PCT	1.10 (1.04–1.15)	0.000 ***
NAGL	1.06 (1.02–1.11)	0.001 **

CI, confidence interval; SOFA, Sequential Organ Failure Assessment; NAGL, neutrophil gelatinase-associated lipocalin; PCT, procalcitonin. *p* value: ** 0.01–0.001; *** < 0.001.

## Data Availability

All data generated or analyzed during this study are included in this published article.

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
