# Peer review of "Th17/Regulatory T-Cell Imbalance and Acute Kidney Injury in Patients with Sepsis"

_jcm, 2022, doi:10.3390/jcm11144027_

Round 1

Reviewer 1 Report

Review jcm-1775549

The paper deals with an interesting topic, but in an incomplete and partial way. The subject of the authors' research is much discussed in the clinical setting.

However, the paper is incomplete in some respects:

- The therapy that patients followed before admission to the ICU is not reported. It would be helpful to know about diabetes mellitus therapy, for exemple (the therapy is only partially mentioned)

- The authors do not deal with alterations in innate immunity. Alterations of innate immunity are a modern topic on the sepsis and AKI scene. Biomarkers such as HMGB-1 and TGFbeta-1 are not mentioned. The authors limited themselves to the study of the lymphocyte component only.

- The citation of biomarkers in AKI and ICU is insufficient (see 26-27-28) and very old references (2009-2010). The value of NGAL as a biomarker has been questioned in subsequent years.   The role of Nephroceck and other biomarkers is much discussed (see recent references).

To publish the paper on JCM it is necessary to review "Discussion" and update the references.

Author Response

Response to Reviewer 1 Comments

Point 1: The therapy that patients followed before admission to the ICU is not reported. It would be helpful to know about diabetes mellitus therapy, for exemple (the therapy is only partially mentioned).

Response 1: Because the sources of patients were wide, including emergency, surgical and medical wards, operating rooms and transferred from other hospitals, etc. Therefore, there was difficult to obtain the treatment measures and achieve quality control for all patients before admission to ICU. However, for patients with infection, timely removal of infection foci, application of antibiotics and fluid therapy are the general principles of treatment.

Point 2: The authors do not deal with alterations in innate immunity. Alterations of innate immunity are a modern topic on the sepsis and AKI scene. Biomarkers such as HMGB-1 and TGFbeta-1 are not mentioned. The authors limited themselves to the study of the lymphocyte component only.

Response 2: The pathogenesis of sepsis remains poorly understood, but is partly attributable to immune over-activation or immunosuppression propagated by dysregulated innate immune responses to lethal infections. Innate immune cells carry various pattern recognition receptors (PRRs) to recognize distinct classes of molecules shared by a group of related microbes, which are collectively termed “pathogen-associated molecular pattern molecules” (PAMPs). The engagement of various PRRs by different PAMPs similarly activates innate immune cells to sequentially release early cytokines (such as tumor necrosis factor (TNF) and interferons (IFNs) and late-acting pro-inflammatory mediators [such as high-mobility group box 1 (HMGB1) and sequestosome 1 (SQSTM1)]. Once released, extracellular HMGB1 can bind many endogenous proteins, thereby modulating divergent innate immune responses to lethal infections. Suppression of HMGB1 may ameliorate inflammation and improve patient outcomes. [Zhu CS, Wang W, Qiang X, et al. Endogenous Regulation and Pharmacological Modulation of Sepsis-Induced HMGB1 Release and Action: An Updated Review[J]. Cells. 2021,10(9):2220. doi: 10.3390/cells10092220. PMID: 34571869.]

The immune system is among the key pathogenic factors in acute kidney injury (AKI). Despite initial skepticism, the bulk of data in the field demonstrates an important role for immune cells in the early pathogenesis of AKI. There is increasing evidence that immune cells are also participating in normal and misguided repair after AKI, with implications for CKD development. There are many important subtypes of leukocytes in AKI, including dendritic cells, macrophages, natural killer (NK) cells, NKT cells, B cells, T cells and neutrophils. T cells play a complex role in AKI and become a hot topic in renal injury and renal repair research.[ Gharaie Fathabad S, Kurzhagen JT, Sadasivam M, Noel S, Bush E, Hamad ARA, Rabb H. T Lymphocytes in Acute Kidney Injury and Repair[J]. Semin Nephrol. 2020 Mar;40(2):114-125. doi: 10.1016/j.semnephrol.2020.01.003. PMID: 32303275.]

In the study, we monitored TNF-a as a biomarker of early inflammatory injury and focused on the  relationship between CD4+ T lymphocyte subsets(Treg cell, Th17 cell and cytokines) and SAKI. It’s truly that we did not monitored level of HMGB-1 or alterations in innate immunity. However, we believe that both innate immunity and T lymphocytes play a role in the development of sepsis and acute kidney injury, and there is an interaction between the two, which requires further explore. Recently, our team confirmed that repressing HMGB1-PTEN signaling can effectively reduce the apoptosis rate of T cells, increase the proliferative activity of T cells, and enhance the function of monocytes in the case of sepsis through animal experiments.[ Zhi D, Zhang M, Lin J,et al. Role of HMGB1-PTEN Signaling in T Lymphocytes and Monocytes Upon Sepsis[J]. Clin Lab. 2022,68(5). doi: 10.7754/Clin.Lab.2021.211024. PMID: 35536087.]This provides evidence and ideas for our follow-up research.

TGF-β can be secreted by Treg cells. TGF-β can directly inhibit T cell activation and play a role in the transformation from Treg cells to Th17 cells. Thanks for your opinion and we planned to detect the level of TGF-β in human serum additionally. We have ordered the kits. At the same time, due to the limitation of experimental subjects in the study, we only analyzed the correlation between Treg/Th17 cells and SAKI. Next, we designed animal and cellular experiments to analyze the causal relationship between Treg/Th17 cells and SAKI. We planned to add TGF-β to cellular experiments.

Point 3: The citation of biomarkers in AKI and ICU is insufficient (see 26-27-28) and very old references (2009-2010). The value of NGAL as a biomarker has been questioned in subsequent years.   The role of Nephroceck and other biomarkers is much discussed (see recent references).

Response 3:Urinary insulin-like growth factor-binding protein (IGFBP-7) and tissue inhibitor of metalloproteinase (TIMP-2) have been proven as the best-performing and have been proposed as a predictive tool for the AKI detection in the critical settings in order to perform an early diagnosis. In September 2014 the test “NephroCheck®” (Astute Medical, San Diego, CA, USA) was approved by the Food and Drug Administration. Kashani K’ study demonstrated that NAGL was weaker than NephroCheck or urinary TIMP-2/IGFBP7, but stronger than plasma cystatin C and urine KIM-1 in predicting acute kidney injury.[ Kashani K, Al-Khafaji A, Ardiles T, et al. Discovery and validation of cell cycle arrest biomarkers in human acute kidney injury[J]. Crit Care. 2013,17(1):R25. doi: 10.1186/cc12503. PMID: 23388612; PMCID: PMC4057242.] Further studies have confirmed that Nephrocheck to predict the risk of acute kidney injury was related to urine flow. As a result, for patients with oliguria or anuria, the predictive ability needs to be judged comprehensively.[ Hahn RG, Yanase F, Zdolsek JH, et al. Serum Creatinine Levels and Nephrocheck® Values With and Without Correction for Urine Dilution-A Multicenter Observational Study[J]. Front Med (Lausanne). 2022,9:847129. doi: 10.3389/fmed.2022.847129. PMID: 35252280.]`

Unfortunately, we did not collected patient's urine, so we could not detect urinary TIMP-2 and IGFBP-7. We deteced serum NAGL as a marker of early kidney injury in the study and got positive results. We reviewed the latest literature, such as Liu’ study and Jia’ study, all of which analyzed NAGL as a predictive biomarker for AKI. [Liu J, Wang Z, Lin J, Li T, Guo X, Pang R, Dong L, Duan M. Xuebijing injection in septic rats mitigates kidney injury, reduces cortical microcirculatory disorders, and suppresses activation of local inflammation[J]. J Ethnopharmacol. 2021,276:114199. doi: 10.1016/j.jep.2021.114199. PMID: 33989736.][ Jia J, Gong X, Zhao Y, Yang Z, Ji K, Luan T, Zang B, Li G. Autophagy Enhancing Contributes to the Organ Protective Effect of Alpha-Lipoic Acid in Septic Rats[J]. Front Immunol. 2019,10:1491. doi: 10.3389/fimmu.2019.01491. PMID: 31333648.] Therefore, we think it is feasible to use NAGL as a biomarker for AKI.

We updated the references of biomarkers in AKI and ICU and enriched "Discussion" of the manuscript as suggested.

Reviewer 2 Report

Thank you for allowing me the opportunity to review this study of  Zhou et al. This is an interesting study on the predictive value oh Th17/Treg on acute kidney injury  of AKI in sepsis patients.

I have some remarks to make to clarify the method of the study:

- The authors must specify the moment of the blood sampling, in the text, it's "after enrollment or diagnosis of SAKI", which is not the same?

- Acute kidney injury before inclusion was not an exclusion criterion, how did the authors deal with those patients?

-Several data in the literature suggest that SARS-Cov2 infection lead to increased inflammatory, often more than bacterial infections, were there any COVID patients in the cohort?

-I think it is more usual to use NGAL than NAGL as an abbreviation for Neutrophil gelatinase-associated lipocalin 

Author Response

Response to Reviewer 2 Comments

Point 1: The authors must specify the moment of the blood sampling, in the text, it's "after enrollment or diagnosis of SAKI", which is not the same?

Response 1: Peripheral venous blood samples were collected immediately after enrollment. When the patients were diagnosised of SAKI, they were enrolled in the study. So the moment of the blood sampling "after enrollment or diagnosis of SAKI" in the manuscript is a duplicate description. We have revised the description as “ Peripheral venous blood samples were collected immediately after enrollment.” in Line 168-169, Page 7.

Point 2: Acute kidney injury before inclusion was not an exclusion criterion, how did the authors deal with those patients?

Response 2: We screened patients with sepsis in the study. As long as the adult septic patients did not meet the exclusion criteria, we performed enrollment screening. In fact, some patients with sepsis already developed acute kidney injury when they were admitted to the ICU. We enrolled this part of patients and divided them into SAKI group. In the part of “Discussion”, we described as “Third, most of the patients had developed sepsis-induced acute kidney injury at the time of enrollment, and the courses of the disease were not entirely consistent.” (seen in Line 461-463, Page 25).This may affect the results, which is one of the limitations of this study.

Point 3:Several data in the literature suggest that SARS-Cov2 infection lead to increased inflammatory, often more than bacterial infections, were there any COVID patients in the cohort?

Response 3: Because our department does not involve COVID patients, there was no COVID patients in the cohort. We mentioned it in Line 216, Page 9.

Point 4: I think it is more usual to use NGAL than NAGL as an abbreviation for Neutrophil gelatinase-associated lipocalin. 

Response 4: NAGL is really better known than ‘Neutrophil gelatinase-associated lipocalin”.We applied “Neutrophil gelatinase-associated lipocalin” a total of 13 times in the manuscript. We used “Neutrophil gelatinase-associated lipocalin” when “NAGL” appeared for the first time in the article and the abbreviation below the tables or figures(9 times) as routine. When we reviewed the literatures, we also saw the full name of NAGL as reference [26] and [28].

Round 2

Reviewer 1 Report

The authors responded sufficiently to the observations.